# Evaluation of the influenza-like illness sentinel surveillance system: A national perspective in Tanzania from January to December 2019

**Vulstan James Shedura**[1]*, **Ally Kassim Hussein**[2], **Salum Kassim Nyanga**[3], **Doreen Kamori**[1], **Geofrey Joseph Mchau**[4]

**1** Muhimbili University of Health and Allied Sciences, School of Public Health and Social Sciences (SPHSS), Dar es Salaam, Tanzania, **2** Tanzania Field Epidemiology and Laboratory Training Program (TFELTP), Dar es Salaam, Tanzania, **3** National Public Health Laboratory, Dar es Salaam, Tanzania, **4** Tanzania Food and Nutrition Centre (TFNC), Dar es Salaam, Tanzania

\* vulstanshedura@gmail.com

## Abstract

### Background

The World Health Organization (WHO) recommends periodic evaluations of influenza surveillance systems to identify areas for improvement and provide evidence of data reliability for policymaking. However, data on the performance of established influenza surveillance systems are limited in Africa, including Tanzania. We aimed to assess the usefulness of the Influenza surveillance system in Tanzania and to ascertain if the system meets its objectives, including; estimating the burden of disease caused by the Influenza virus in Tanzania and identifying any circulating viral strains with pandemic potential.

### Methodology

From March to April 2021, we collected retrospective data through a review of the Tanzania National Influenza Surveillance System electronic forms for 2019. Furthermore, we interviewed the surveillance personnel about the system's description and operating procedures. Case definition (ILI-Influenza Like Illness and SARI-Severe Acute Respiratory Illness), results, and demographic characteristics of each patient were obtained from the Laboratory Information System (Disa*Lab) at Tanzania National Influenza Center. The United States Centers for disease control and prevention updated guidelines for evaluating public health surveillance systems were used to evaluate the system's attributes. Additionally, the system's performance indicators (including turnaround time) were obtained by evaluating Surveillance system attributes, each being scored on a scale of 1 to 5 (very poor to excellent performance).

### Results

A total of 1731 nasopharyngeal and oropharyngeal samples were collected from each suspected influenza case in 2019 from fourteen (14/14) sentinel sites of the influenza surveillance system in Tanzania. Laboratory-confirmed cases were 21.5% (373/1731) with a

**Data Availability Statement:** All relevant data are within the manuscript and its Supporting Information files (S4-File).

**Funding:** The author(s) received no specific funding for this work.

**Competing interests:** The authors have declared that no competing interests exist.

predictive value positive of 21.7%. The majority of patients (76.1%) tested positive for Influenza A. Thirty-seven percent of patients' results met the required turnaround time, and 40% of case-based forms were incompletely filled. Although the accuracy of the data was good (100%), the consistency of the data was below (77%) the established target of $\geq$ 95%.

## Conclusion

The overall system performance was satisfactory in conforming with its objectives and generating accurate data, with an average performance of 100%. The system's complexity contributed to the reduced consistency of data from sentinel sites to the National Public Health Laboratory of Tanzania. Improvement in the use of the available data could be made to inform and promote preventive measures, especially among the most vulnerable population. Increasing sentinel sites would increase population coverage and the level of system representativeness.

## Introduction

*Influenza* is an acute viral respiratory tract disease characterized by fever, headache, myalgia, prostration, coryza, sore throat, and cough. Cough is often severe and prolonged, with recovery in 2–7 days [1]. The World Health Organization (WHO) estimates that seasonal influenza causes 3 to 5 million cases of severe influenza disease and may result in 290 000–650 000 deaths yearly due to respiratory diseases alone [2]. In 2008, Tanzania initiated the National Influenza Sentinel Surveillance System (NISSS), whereby on July 4, 2009, the first case was reported [3], and since then, epidemics have been reported and monitored. Currently, the documented prevalence of seasonal influenza in Tanzania is 8.0%, and according to the first surveillance evaluation done in 2010 (30 months after the initiation of the surveillance system), patients increased during the cold and rainy seasons [4].

Global influenza surveillance, coordinated by WHO under the Global Influenza Surveillance and Response System (GISRS) Network, is key to monitoring global trends of seasonal influenza virus circulation, guiding strain selection for annual influenza vaccine composition, monitoring the acquisition of resistance to antiviral drugs, detecting the emergence of influenza viruses with pandemic potential, and monitoring the spread and impact of pandemic influenza viruses [5].

The GISRS network was initiated in 1952 by the WHO as a response strategy against influenza disease [3]. WHO recommends using standard case definitions and procedures for global influenza surveillance among outpatients and inpatients and periodic comprehensive evaluations of established surveillance systems, beginning 1–2 years after implementation [3].

The NISSS in Tanzania operates through an organizational structure that involves the sentinel sites, the National Public Health Laboratory (NPHL), the epidemiology section of the Tanzania Ministry of Health (MoH), and collaborative surveillance partners (WHO and Centers for disease control and prevention (CDC) (see S1 Fig).

Sentinel sites are hospitals or health centers, and each has a surveillance focal person who coordinates the functioning of the surveillance activity, linking and reporting to NPHL and the Epidemiology section at the MoH. The main functions of a sentinel site are to enroll patients for surveillance, collect epidemiological and laboratory data, perform data entry and simple analysis and submit collected Case-Based Forms (CBF) and laboratory specimens to

the Epidemiology section of the MoH and NPHL, respectively [3]. NPHL, as a National Influenza Center (NIC) of Tanzania, has the role of supporting laboratory capacity for the surveillance system at the central level (MoH) and sentinel sites [3]. The Epidemiology Section of the MoH is responsible for the overall coordination, staff training, response, procurement of equipment and consumables, monitoring and evaluation, and providing feedback to all levels. They are also primarily responsible for data management, analysis, and dissemination of reports to stakeholders [3].

The overall responsibilities of collaborative partners are to contribute to technical assistance and provide resource support for running surveillance activities [3]. The primary purpose of implementing an influenza sentinel surveillance system includes: determining the characteristics of circulating Influenza viruses as well as detecting and ascertaining any new strains with pandemic potential [3].

Although influenza sentinel surveillance has been established in several African countries, data about the performance of established surveillance systems are limited on the continent and especially in Tanzania [6–8]. Such evaluations enable countries to assess the performance of their surveillance systems, identify areas for improvement and provide evidence of data reliability for policymaking and public health interventions as well as compliance with international surveillance standards. This study aimed to evaluate the NISSS in Tanzania by determining whether the NISSS complies with its objectives stated in the protocol for the national influenza sentinel surveillance system [3]. It also evaluated the system's attributes and usefulness and provided relevant recommendations for improvement. Findings from this evaluation will help to improve the performance of the influenza surveillance system in Tanzania and other similar resource-limited settings.

## Materials and methods

We described the attributes and usefulness of the NISSS of Tanzania using a descriptive cross-sectional survey. We extracted and evaluated routinely recorded ILI and SARI data between January and December 2019.

### Population under surveillance

The population under NISSS involves patients enrolled from the fourteen sentinel facilities in Kigoma, Dodoma, Mwanza, Manyara, Arusha, Mtwara, Dar es Salaam regions, and Zanzibar. These sentinel sites are Bombo regional hospital, Bububu hospital, Dodoma regional referral hospital (DRRH), Hydom Lutheran hospital, International School of Tanganyika Clinic center (IST Clinic), Istiqaama hospital, Kibondo district hospital, Mbalizi hospital, Meru district hospital, Mt. Meru regional referral hospital, Mwananyamala regional referral hospital (MRRH), Mzinga hospital, Sekou-Toure regional referral hospital and St. Benedict's Ndanda hospital (SBNH).

### Inclusion criteria

The data of all influenza suspects detected by either Influenza-like illness (ILI) or Severe acute respiratory infections (SARI) case definition enrolled in the surveillance system and therefore included in the 2019 influenza data set (electronic forms) were used for analysis. We defined an *ILI case* as " a person with an acute respiratory infection with the measured fever of $\geq$ 38 C˚ and cough, with onset within the last ten days". *SARI case was defined as* "a person with an acute respiratory infection with a history of fever or measured fever of $\geq$ 38 C˚, and cough; with onset within the last ten days, and requires hospitalization" [3].

### Exclusion criteria

We excluded patients whose data were missing important information such as age, case definition type, name of sentinel site, and influenza results. In addition, stakeholders who were unwilling to participate in the interview were not involved in this study.

### Study area

We evaluated the MoH epidemiology section in Dodoma, NPHL in Dar Es Salaam, and three selected sentinel sites: SBNH in the Mtwara region, DRRH in the Dodoma region, and MRRH in the Dar es salaam region. The sites were selected purposely due to location, different levels of care, and relatively higher number of patients enrolled in the NISSS in Tanzania compared to other sentinel sites.

### Specimen collection and laboratory procedures

The biospecimens used in the NISSS were nasopharyngeal and oropharyngeal swabs. A nasopharyngeal and an oropharyngeal swab sample for all patients who met the case definition (ILI/SARI) were collected from each sentinel site. In the sites, both swabs were placed into a single cryovial containing a viral transport medium. Cryovials containing specimens were immediately refrigerated and transported to the NIC in Dar Es Salaam via courier services. Upon receipt of specimens at NIC, the specimens were immediately stored at -80°C freezer before testing. The data for all samples received at NIC were entered into the Laboratory information system. Samples were tested for influenza A and influenza B viruses by real-time reverse-transcription polymerase chain reaction (rRT-PCR), using the CDC protocol for detection and characterization of seasonal influenza virus and 2009 pandemic influenza A virus subtype H1N1pdm09) as well as the WHO manual for the laboratory diagnosis and virological surveillance of influenza [4]. The RNA extraction from 140-μl aliquots of each specimen was performed using a QIamp viral RNA Minikit (Qiagen, Germany) according to the manufacturer's instructions. A one-step rRT-PCR was performed using the AgPath kit (Applied Biosystems, Carlsbad. CA). Appropriate negative and positive control specimens were run alongside each reaction. The results were recorded as cycle threshold (CT) values. When controls met stated requirements, any influenza A or B virus results with a CT value of $\leq$ 39.9 as positive were recorded, and those with a CT reading of $\geq$40 were recorded as negative [5]. All specimens positive for influenza A virus were subtyped for seasonal H1, H3, and H1N1 pdm09, using rRT-PCR. Specimens positive for influenza A virus by rRT-PCR but not sub-typable were sent to the WHO influenza collaborating center for further antigenic characterization.

### Data collection procedure and surveillance evaluation techniques

Data on ILI and SARI from all fourteen sentinel facilities in Tanzania were extracted and abstracted from the NIC database (Disa*Lab) into Microsoft (MS) Excel® version 2019 format. Additionally, we selected three sentinel sites based on their location (southern, eastern, and central parts of the country), the volume of patients enrolled, and the different levels of care they provided (primary to tertiary hospital), and obtained permission for site visits.

The researchers interviewed all personnel directly involved in National Influenza Sentinel surveillance and also partook in surveillance activities in each of the selected sentinel sites while observing practices. The staff members were interviewed using a structured questionnaire and guided interviews on their knowledge of surveillance activities (see S1 File). We used the tally method to capture data on the completeness of case-based forms of influenza at NIC.

We evaluated the surveillance system's attributes using CDC guidelines for evaluating public health surveillance systems (see S2 File). We used the unique serial numbers on case investigation forms for each sentinel site to retrieve the randomized forms for examination. In addition, we collected and reviewed Influenza surveillance registers at the sentinel sites visited.

The system's attributes evaluated include usefulness, simplicity, flexibility, representativeness, timeliness, data quality, acceptability, sensitivity, predictive value positive, and stability of the NISSS. We evaluated these attributes as follows:

*The usefulness of the surveillance system* entails describing how the system provides information that is useful for public health authorities and communities (see S2 File). We assessed the usefulness of the Influenza surveillance system among the stakeholders from all selected sentinel sites through questionnaires and observation methods. The stakeholders provided the use status of the NISSS, and we assessed the utilization of the data through observation of the daily operations. In evaluating this attribute, the periodic reports and bulletin, case definition (SARI, ILI), interview guide and questionnaires, and case- registers were used to obtain this information. We observed the proportion of surveillance staff that reported that they regularly receive the monthly influenza bulletin, availability of annual influenza reports, identification and sharing of circulating seasonal influenza strains, and the outbreaks detected over the pre-established threshold. *The simplicity of the surveillance system* refers to both structure and ease of operation of a public health surveillance system (see S2 File). We assessed the simplicity of the NISSS through questionnaires and observation. We assessed the proportion of samples collected from admitted SARI cases at the sentinel sites as part of the NISSS operations.

Furthermore, in assessing this attribute, we observed the data collection methods and shipment and stakeholders' perceptions of the NISSS operations using the structured questionnaire.

*The surveillance system's flexibility is the ability* to change information needs or operating conditions with little additional time, personnel, or allocated funds (see S2 File). We assessed the flexibility of NISSS through observation and review of reports and charts used by surveillance stakeholders. We also assessed the changes in standard case definition with time, new illnesses revealed, and the ability to include or detect other viral respiratory pathogens surveyed in NISSS. *Representativeness of the surveillance system* is the occurrence of a health-related event over time and its distribution in the population by place and person (see S2 File). In this evaluation, we assessed the representativeness of the NISSS through a review of periodic reports produced and the NISSS protocol in Tanzania. We checked the reports on the distribution of cases in all age categories and the selection criteria of the sentinel sites and geographical coverage.

*The timeliness of the surveillance system* reflects the speed between steps in a public health surveillance system (see S2 File). The timeliness of the NISSS relates to the Turnaround Time (TAT) of the laboratory reports. TAT refers to the time elapsed from receiving samples in the Laboratory to releasing results (see S2 File). We assessed the timeliness of NISSS through a review of the TAT of laboratory reports since this can impact the timeliness of all operations of the entire system. We obtained the proportion of the timely reports released within pre-established TAT (7 days). *Data quality of the surveillance system* refers to the completeness and validity of the data recorded in the public health surveillance system (see S2 File). In evaluating this attribute, we assessed the completeness of the reports and case-based forms' accuracy and consistency of data yielded from the NISSS. In assessing the completeness of NISSS, we obtained the proportion of complete filled reports and case forms. We obtained data accuracy information through the assessment of laboratory quality assurance protocol (internal quality control and external quality assessments). Furthermore, we evaluated data consistency by comparing the data of the same patients between sentinel sites and NIC in real time.

*Acceptability of the surveillance system* is the willingness of persons and organizations to participate in the surveillance system (see S2 File). In assessing this attribute, we interviewed

the stakeholders at the sentinel sites, NIC, MoH, and collaborating partners (WHO and CDC) of the NISSS in Tanzania. In addition, we observed the general performance of other attributes. *The sensitivity of the surveillance system* (at the level of outbreak detection) is the ability of the surveillance system to detect an outbreak. In evaluating the sensitivity of the NISSS, we observed the reports and bulletin to see the number of cases obtained against the pre-established threshold in the NISSS protocol in Tanzania. We obtained the proportion of the outbreaks detected against the target.

*The predictive value Positive (PVP)* of the surveillance system refers to the proportion of reported cases with the health-related event under surveillance (see S2 File). We performed the PVP calculation using a formula: True positive/ (True positive + False negative). The "True positive" of the surveillance system are those patients who met the standard case definition of either SARI or ILI, depending on their clinical presentations, and therefore enrolled in the surveillance system, and their PCR results showed positive results of the influenza Virus A or Influenza Virus B. However, the other category of "False positive" of the surveillance system is those patients who met the standard case definitions (SARI/ILI) and enrolled in the surveillance system, but their PCR results showed no detection of the Influenza virus.

*The stability of the surveillance system* is the reliability and availability of the public health surveillance system (see S2 File). In assessing the stability of the NISSS, we interviewed the stakeholders and reviewed the system protocol to observe the funding mechanism of the NISSS including the number of internal and external funding sources.

We measured the performance of the NISSS using a scale from 1 to 5 to provide a score for each indicator with a percentage value from [0–20% [score 1 (very poor performance); [20–40% [score 2 (poor performance); [40–60% [score 3 (moderate performance); [60–80% [score 4 (good performance); [80–100% [score 5 (excellent performance). These scores are based on the consensus opinions of surveillance experts: virologists, public health specialists, and epidemiologists working at the NISSS.

## Data processing and analysis

Data were entered, cleaned, and stored in Microsoft (MS) Excel® version 2019, and we analyzed the system attributes using Microsoft Excel® and Stata® version 15.1. We calculated the frequency and proportions to evaluate the system attributes using predefined indicators described in the NISSS protocol (see S2 File).

## Ethics statement

We evaluated the NISSS within the framework of the Integrated Disease Surveillance and Response matrix implemented by the Tanzania MoH and therefore did not require a formal review by Ethical Review Committees. This study did not involve direct contact with the patients; we based it on the verbal consent obtained from patients before enrolment into the surveillance. We sought and obtained permission from the MoH of Tanzania for the epidemiology section and authorities in the sentinel facilities prior to the commencement of the evaluation process. We anonymized the data before being analyzed to ensure patient confidentiality.

## Results

### Characteristics of the study participants

The Influenza Laboratory electronic forms from the Laboratory information system (Disa*Lab) at NIC showed that; a total of 1731 nasopharyngeal and oropharyngeal swabs

specimens from suspected cases of influenza were collected from January to December 2019, of which 912 (52.7%) were from male participants. The individuals in the surveillance system had a median age of 3 years (Interquartile range (IQR) = 0–28). Forty-one percent (708/1731) of patients were enrolled using the SARI case definition type in the surveillance system. Eight percent (83/1023) of patients who presented with ILI symptoms were confirmed positive for influenza B virus, and only 0.85% (6/708) of these presented with SARI symptoms during the time of enrolment (see Table 1 and Fig 1). Of all samples from ILI cases, 21.5% (373/1731)

**Table 1. Socio-demographic characteristics of the people under the National Influenza Sentinel Surveillance System in Tanzania, 2019.**

| Characteristic | Case definition type | | |
|---|---|---|---|
| **Age (years)** | ILI (n = 1023) | SARI (n = 708) | Total (N = 1731) |
| | n (%) | n (%) | n (%) |
| 0–4 | 384 (37.54) | 569 (80.37) | 953 (55.05) |
| 5–14 | 101 (9.87) | 27 (3.81) | 128 (7.39) |
| 15–24 | 137 (13.39) | 23 (3.25) | 160 (9.24) |
| 25–34 | 174 (17.01) | 18 (2.54) | 192 (11.09) |
| 35–44 | 98 (9.58) | 18 (2.54) | 116 (6.70) |
| 45–54 | 47 (4.59) | 14 (1.98) | 61 (3.52) |
| 55–64 | 53 (5.18) | 15 (2.12) | 68 (3.93) |
| ≥65 | 29 (2.83) | 24 (3.39) | 53 (3.06) |
| Median (IQR) | 18 (2–33) | 1 (0–3) | 3 (0–28) |
| **Sex** | | | |
| Female | 531 (51.91) | 276 (38.98) | 807 (46.62) |
| Male | 481 (47.02) | 431 (60.88) | 912 (52.69) |
| Missing | 11 (1.08) | 1 (0.14) | 12 (0.69) |
| **Sentinel site** | | | |
| Bombo regional hospital | 2 (0.20) | 0 (0.00) | 2 (0.12) |
| Bububu hospital | 184 (17.99) | 0 (0.00) | 184 (10.63) |
| Dodoma regional referral hospital | 6 (0.59) | 50 (7.06) | 56 (3.24) |
| Hydom Lutheran hospital | 47 (4.59) | 38 (5.37) | 85 (4.91) |
| International School of Tanganyika Clinic | 39 (3.81) | 0 (0.00) | 39 (2.25) |
| Istiqaama hospital | 5 (0.49) | 0 (0.00) | 5 (0.29) |
| Kibondo district hospital | 24 (2.35) | 100 (14.12) | 124 (7.16) |
| Mbalizi hospital | 193 (18.87) | 0 (0.00) | 193 (11.15) |
| Meru district hospital | 97 (9.48) | 41 (5.79) | 138 (7.97) |
| Mt. Meru regional referral hospital | 64 (6.26) | 99 (13.98) | 163 (9.42) |
| Mwananyamala regional referral hospital | 170 (16.62) | 277 (39.12) | 447 (25.82) |
| Mzinga hospital | 65 (6.35) | 0 (0.00) | 65 (3.76) |
| Sekou-Touré regional referral hospital | 43 (4.20) | 27 (3.81) | 70 (4.04) |
| St. Benedict's Ndanda hospital | 84 (8.21) | 76 (10.73) | 160 (9.24) |
| **Influenza virus results (rRT-PCR)** | | | |
| Influenza A Detected | 175 (17.11) | 109 (15.40) | 284 (16.41) |
| A(H3N2) | 31 (3.03) | 23 (3.25) | 54 (3.12) |
| A(Unclassified) | 144 (14.08) | 86 (12.15) | 230 (13.29) |
| Influenza B Detected | 83 (8.11) | 6 (0.85) | 89 (5.14) |
| Influenza A and B Not Detected | 757 (74.00) | 591 (83.47) | 1348 (77.87) |
| Failed samples | 8 (80.00) | 2 (0.28) | 10 (0.58) |
| **Turnaround time (TAT)** | | | |
| ≤ 7 days | 302 (29.52) | 331 (46.75) | 633 (36.57) |
| >7 days | 721 (70.48) | 377 (53.25) | 1098 (63.43) |

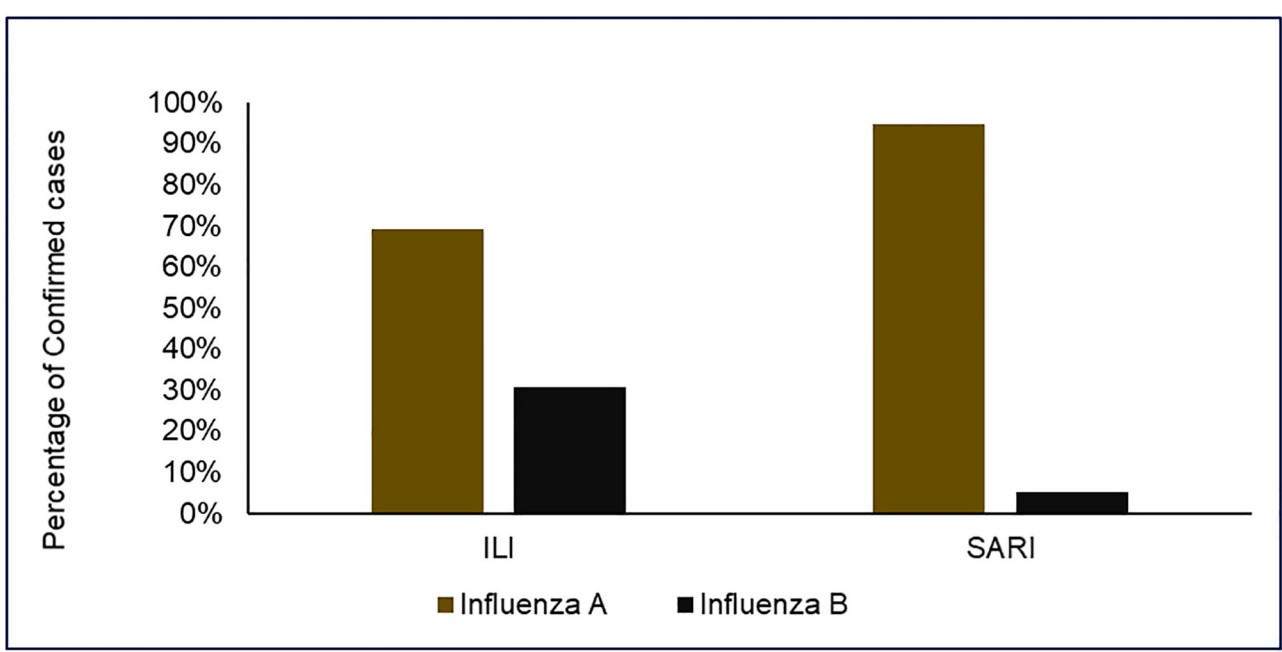

**Fig 1. Distribution of influenza types in Tanzania according to case definition used among confirmed cases of influenza from January to December 2019 (N = 373).**

were laboratory-confirmed influenza cases. Most samples were collected from MRRH [25.8% (447/1731)], and the least number of samples were from Bombo regional hospital [0.1% (2/ 1731)]. Most laboratory results of patients under surveillance [63.4% (1098/1731)] did not meet the required turnaround time (TAT) of ≤7 days for ≥90% of all patient results (see Table 1). The TAT of the patients' results captured was less than 50% of the target in most sentinel sites, regardless of their distance from the NIC (see Fig 4).

## Usefulness (Utility) of influenza surveillance system

We assessed the usefulness of the Influenza surveillance system among the stakeholders from all selected sentinel sites, and 80% (10/12) agreed on the importance of NISSS in Tanzania. Furthermore, the surveillance system provided valuable data to all stakeholders via periodic reports, including weekly, monthly and annual influenza surveillance bulletins. Additionally, national surveillance data allowed the calculation of the disease burden and showed the weekly distribution of influenza cases in the country (see Fig 2). The mean score for the utility was 5 (see Table 3).

## Simplicity

Only 21.4% (600/2803) of all patients admitted to sentinel sites (SBNH, DRRH, and MRRH) enrolled by SARI standard case definition their oropharyngeal and nasopharyngeal samples were collected for laboratory testing (see Table 2). Furthermore, we also observed the use of manual registers at the sentinel sites, which may have influenced the delay of surveillance activities, compromising the data accuracy. These data from selected sites show that the surveillance system is complex due to its data collection method. The mean score for simplicity was 3 (see Table 3).

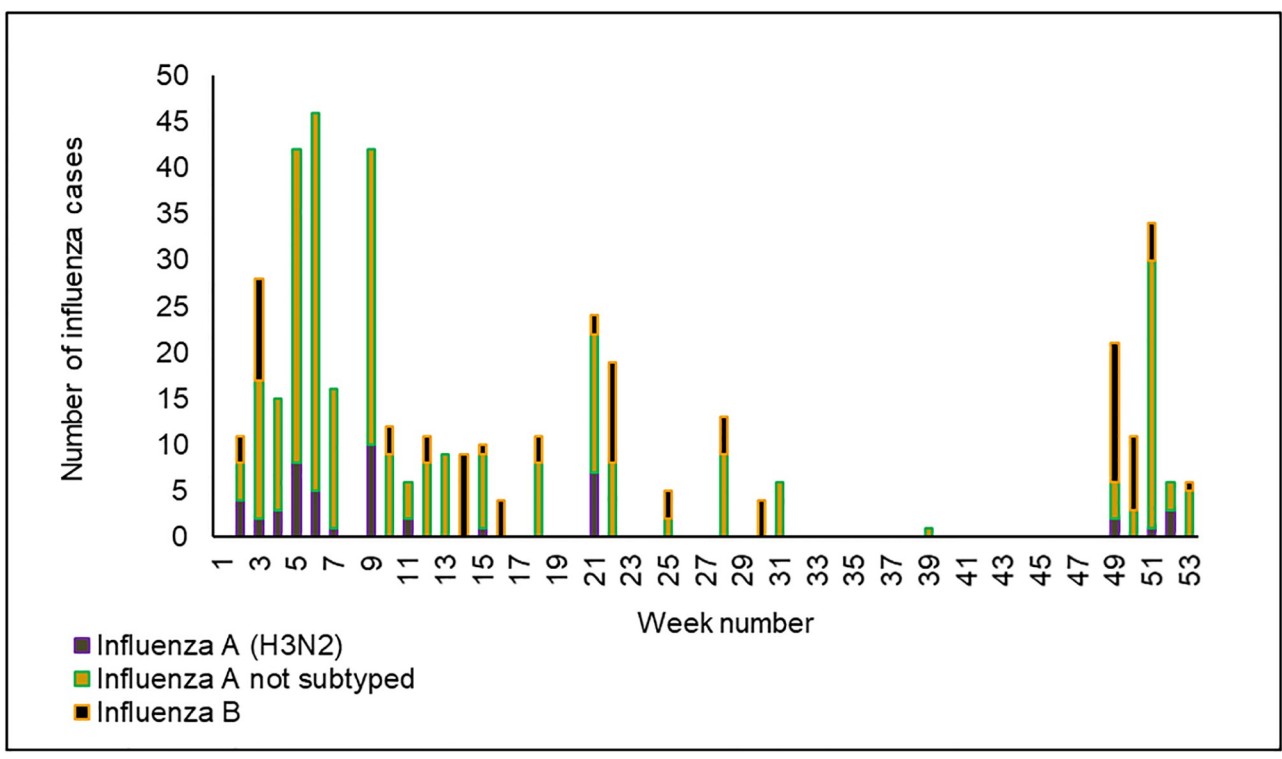

**Fig 2. Weekly distribution of influenza cases in Tanzania from January to December 2019 (N = 373).**

## Data quality

In this attribute, we examined the data completeness, accuracy, and data consistency, and the results were as follows:

## Data consistency

The overall data consistency between sentinel sites and NPHL was 77% (663/861). We observed the highest discrepancy in samples sent from MRRH to NPHL with a standard deviation of 29.0 (see S3 File). The mean score for data consistency was 4 (See Table 3).

**Table 2. The distribution of the SARI patients under the National Influenza Sentinel Surveillance System and the average samples collected per study sentinel sites in Tanzania, 2019.**

| | Sentinel site | | | |
|---|---|---|---|---|
| | **SBNH** | **DRRH** | **MRRH** | **Total** |
| **P1** | 1186 | 79 | 1538 | 2803 |
| **ST** | 122 (10%) | 79 (100%) | 399(26%) | 600 (21%) |
| **AC** | 10 (13%) | 7 (9%) | 33 (41%) | 50 (63%) |

**Key: P1:** Admitted patients who fulfilled the influenza case definition for SARI; **ST**: Total number of samples from SARI admitted patients from January to December 2019; **AC**: Average target set for a sample collection from January to December 2019 per month (80 samples per month).

**Table 3. Findings from the evaluation of the National Influenza Sentinel Surveillance System in Tanzania, 2019.**

| System attribute | Indicators | Score (%) | Mean score (%) | Target score (%) |
|---|---|---|---|---|
| **Utility/Usefulness** | Importance of the surveillance system | 5 (80) | 5 (92) | 5 (≥80) |
| | Provided useful data | 5 (90) | | |
| | The proportion of surveillance staff that reported that they regularly receive the monthly Influenza bulletin | 5 (100) | | |
| | Annual Influenza report | 5 (100) | | |
| | Identification and sharing of circulating seasonal influenza strains | 5 (80) | | |
| | Outbreaks detected over the pre-established threshold | 5 (100) | | |
| **Simplicity** | The proportion of samples collected from admitted SARI patients [N: Number of samples collected from admitted SARI patients; D: Total number of patients that met SARI case definition] | 2 (21) | 3 (57) | 5 (≥80) |
| | Data collection and shipment | 3 (50) | | |
| | Perception of stakeholders on the simplicity of the system | 5 (100) | | |
| **Flexibility** | Changes in standard case definition | 5 (100) | 5 (100) | 5 (≥80) |
| | New illnesses revealed by the system | 5 (100) | 5 (100) | |
| | Inclusion of other viral respiratory pathogens surveyed with the influenza surveillance system | | | |
| **Representativeness** | The total number of patients captured | 3 (50) | 4 (64) | 5 (≥80) |
| | Number of sentinel sites involved | 4 (60) | | |
| | Geographical coverage of the sentinel sites | 3 (44) | | |
| | Inclusion of all age groups in NISSS | 5 (100) | | |
| **Timeliness** | The proportion of timely reporting of results [N: Number of results reported within required TAT (within 1 week); D: Total number of results reported at a time period]. | 3 (55) | 3 (55) | 5 (≥90) |
| **Data quality:** | | | | |
| **i. Completeness** | Percentage of blank forms [N: Number of incomplete filled forms; D: Total Number of evaluated forms] | 3 (40) | 3 (40) | 1 (<10) |
| | Percentage of complete filled case-based forms/ reports [N: Number of complete filled forms: Total number of evaluated forms case-based forms and/or periodic reports]. | 3 (60) | 3 (60) | 5 (≥90) |
| **ii. Data accuracy** | % EQA performance [N: Number of EQA panels with performance ≥90%; D: Total number of EQA panels evaluated]. | 5 (100) | 5 (100) | 5 (≥90) |
| | % Of IQC passed within a selected time period [N: Total number of IQC passed; D: Total number of IQC evaluated]. | 5 (100) | 5 (100) | 5 (100) |
| **iii. Data consistency** | % Consistency between samples sent from selected sentinel sites to NPHL to those received at NPHL [Proportion of the difference in the number of samples sent from sentinel sites to NPHL to the number of the same samples received at NPHL] | 4 (77) | 4 (77) | 5 (≥95) |
| **Acceptability** | Performance observation from other attributes | 3 (50) | 4 (75) | 5 (≥95) |
| | Perception of stakeholders on accepting the system [N: Number of stakeholders accepting the system; D: Total number of stakeholders evaluated via interview]. | 5 (100) | | |
| **Sensitivity** | The proportion of the number of outbreaks identified by the system [N: Number of outbreaks detected by the system; D: Total number of outbreaks that occurred in a time period]. | 5 (100) | 5 (100) | 5 (≥90) |
| **Predictive value Positive (PVP)** | The proportion of cases correctly picked by the surveillance system (TP); the number of healthy individuals picked as cases by the system (FP). [N: Number of patients truly identified by the system (TP); D: Total number of patients identified by the surveillance system (TP+FP)]. | 2 (21) | 2 (21) | 5 (≥90) |
| **Stability** | The number of funding sources of the system presence of internal and/or external funding mechanisms. | 4 (80) | 4 (80) | 5 (≥80) |

**Key:** Surveillance performance description: a scale from 1 to 5 was used to provide a score for each indicator with a percentage value as follows:

[0–20% [score 1 (very poor performance); [20–40% [score 2 (poor performance); [40–60% [score 3 (moderate performance); [60–80% [score 4 (good performance); [80–100% [score 5 (excellent performance). (N: Numerator, D: Denominator).

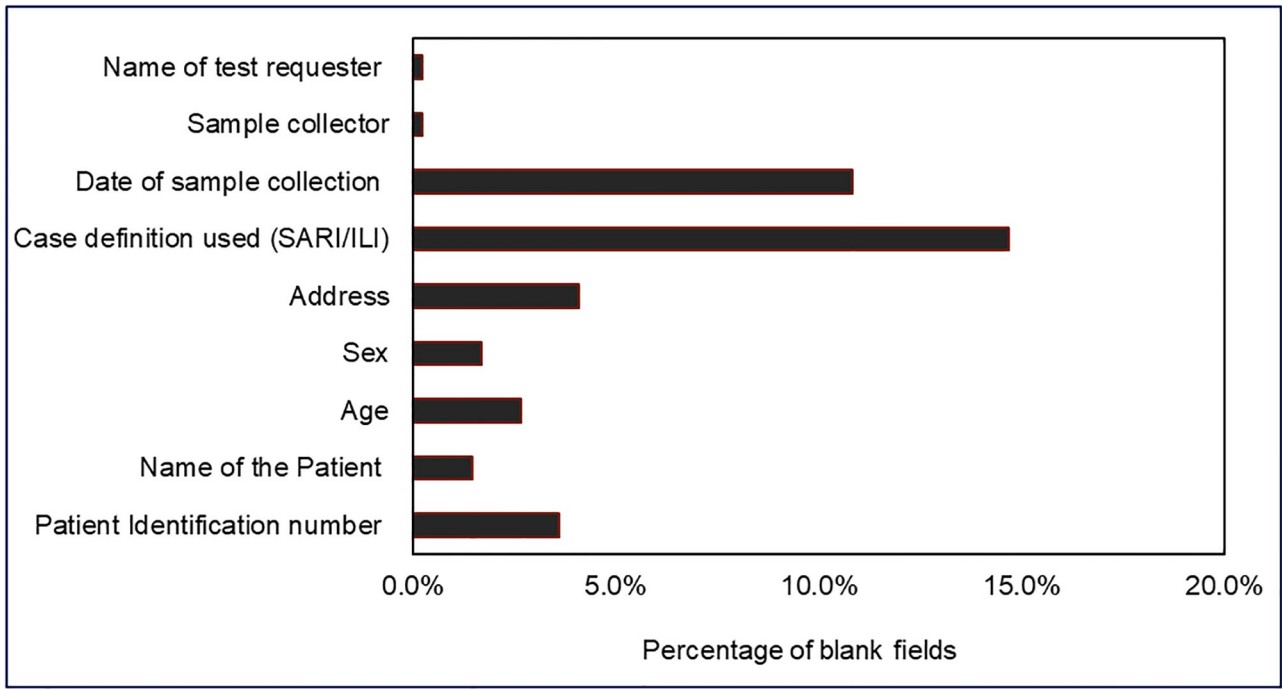

**Fig 3. The proportion of incompleteness of selected parameters on influenza case-based forms used by the National Influenza Sentinel Surveillance System in Tanzania, 2019 (N = 415).**

## Data completeness

We observed that 60% (164/415) of the forms were incompletely filled. We saw that most of the blanks on these forms were in the case definition type (15%) as well as the date of sample collection (11%) fields (see Fig 3). The mean score for data completeness was 3 (see Table 3).

## Data accuracy

Through document review, we observed that; Internal Quality Control (IQC) was always done in each run of the samples and also passed External Quality Assessment (EQA) with 100% concordance with WHO standards. Furthermore, NPHL had an average performance of 100% in all EQA panels. The mean score for data accuracy was 5 (Table 3).

## Flexibility

The system is flexible after standard case definitions were revised in 2019 to comply with WHO's standards and prove to work through the system [9]. Also, the system, through influenza Laboratory testing protocol, allows the investigation of other viral respiratory infections apart from influenza. The mean score for flexibility was 5 (see Table 3).

## Representativeness

Sentinel sites were selected representatively (not randomly) based on population density, climate, and socio-diversity to estimate the country's disease burden. We captured data from the 14 sentinel sites found in eight regions of Tanzania. Furthermore, the distribution of people

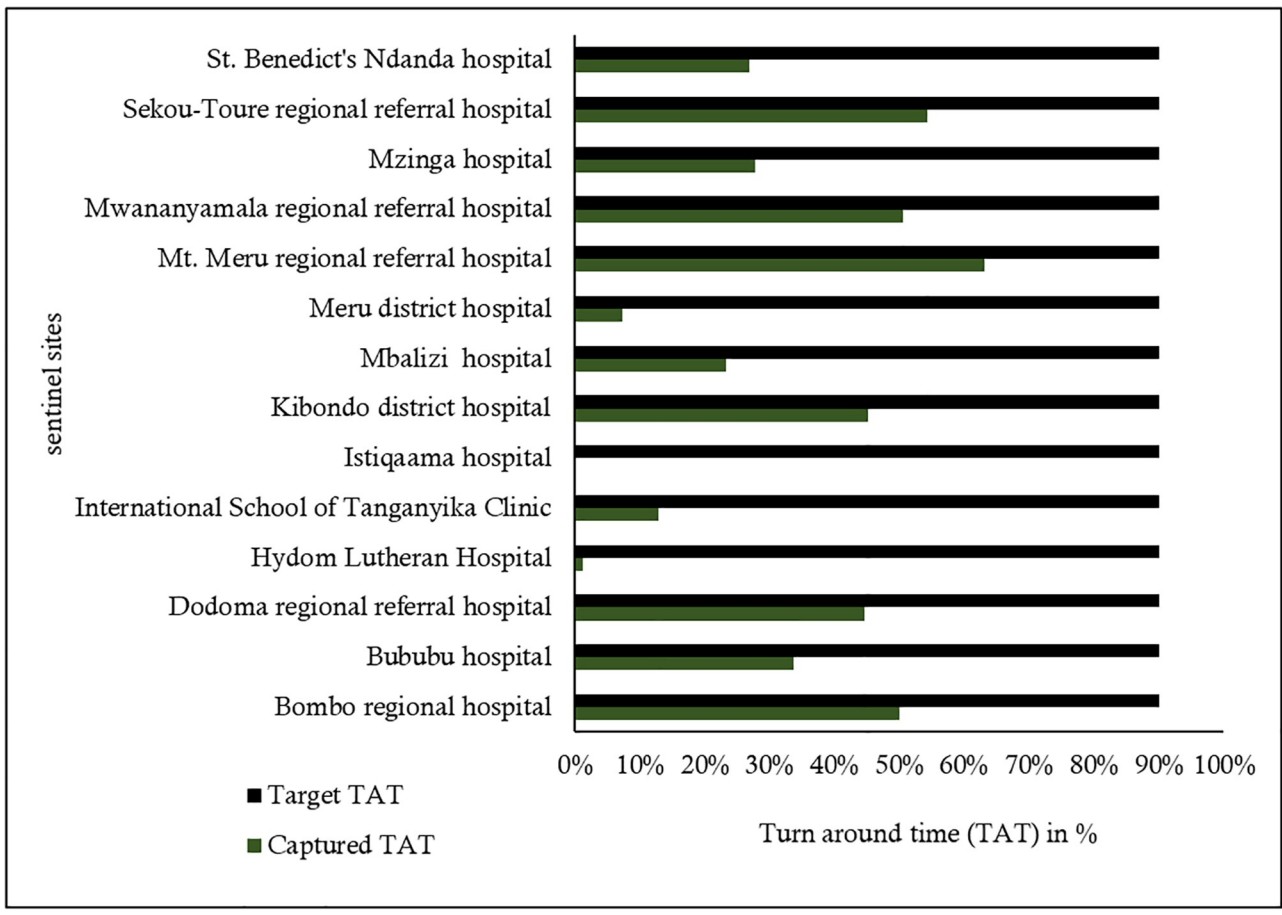

**Fig 4. Turnaround coverage among the sentinel sites of the National Influenza Sentinel Surveillance System (NISSS) in Tanzania, January to December 2019.**

under surveillance involved all age groups (see Table 1). The mean score for representativeness was 4 (see Table 3).

## Timeliness

Among all patients' samples tested for rRT-PCR influenza test, only 36.6% (633/1731) met the standard TAT (Target is ≥90%). Furthermore, 57.1% (8/14) of all sentinel sites had less than 50% (Target ≥90%) of patient samples met the standard TAT (see Fig 4). The mean score for timeliness was 3 (see Table 3).

## Acceptability

The interviewed stakeholders accepted the operations of the NISSS by 100% (12/12). The mean score for acceptability of the NISSS was 4 (see Table 3).

## Sensitivity

The NISSS could detect cases of influenza and could confirm several outbreaks of influenza in different regions of Tanzania. The mean score for the sensitivity of the NISSS was 5 (see Table 3).

### Predictive value positive (PVP)

The number of patients enrolled in influenza surveillance who tested positive for influenza by rRT-PCR in 2019 was 373, and the number of those enrolled but had negative rRT-PCR results was 1348. Therefore, PVP was 21.7% (373/1721). The mean score for PVP of the NISSS was 2 (see Table 3).

### Stability

The system utilizes employed staff at all levels. The system is financially stable since WHO and the United States government fund it through the CDC and the MoH of Tanzania. The mean score for the stability of the NISSS was 4 (see Table 3).

## Discussion

This study assessed the NISSS in Tanzania to provide comprehensive data on strengths and weaknesses and provide suitable recommendations for improvement. The observed distribution of the influenza disease by sex, influenza type, and case definition type, from the NISSS in Tanzania, has not shown any significant difference from the study done by Mmbaga *et al.* in 2012 [4].

This evaluation has shown that the NISSS in Tanzania has moderate overall performance (with a mean score of 3 to 4). Although the PVP was low [21.7% (target ≥90%)], this is relatively higher compared to the PVP of 7.9% obtained from the previous evaluation report on 30 months of evaluation after the establishment of the NISSS in Tanzania [4]. The observed inconsistency of data from the sentinel sites to the National Influenza Laboratory (that is, 23% data discrepancy compared with the target of <5%) is most likely due to errors during documentation at the sentinel sites.

The observed higher TAT [only 37% (target ≥90%)] is likely due to prolonged registration time at NPHL, which may be due to a shortage of human resources, which could hasten the registration process of case-based forms in LIS before testing. Although all sentinel sites did not meet the target TAT, their distance from NIC may have contributed to the delay of patients' results, and thus, we suggest further investigation that will address the TAT captured in every sentinel site and their distance from NIC. The timeliness of the NISSS in Tanzania may have been affected by the geographical distribution of sentinel sites and the simplicity of data collection and reporting methods.

An electronic information management system (from the sentinel sites level to the NIC) was one of the most critical recommendations proposed by surveillance staff to improve data completeness, quality monitoring, and timeliness. The electronic system would facilitate real-time data sharing at all levels involved in the NISSS [10]. The evaluation of the syndromic influenza sentinel surveillance system in Madagascar between 2009 and 2014 showed that by using mobile phones and texting for the transmission of daily aggregated data, the simplicity of the system was strengthened, improving the completeness, quality and timeliness of the data as well as acceptability of the system to the sentinel site staff [11]. This is in line with the recommendation that a revised data management approach, which offers the advantage of automation and a user-friendly interface between the sentinel sites and the NIC, would ultimately improve the acceptance and utility of the surveillance system.

Although system representativeness was moderate (64%) due to the involvement of all age categories in the surveillance, the number of sentinel sites are still inadequate (only 14 sentinel sites available) to cover the entire population of Tanzania and thus, increasing more sentinel sites would have increased the coverage and the level of system representativeness. Having a lower PVP suggests the need for the National Influenza Technical Working Group (NITWG)

to plan for reviewing the current case definition to improve the PVP and thus ensure that the NISSS enrolls the actual cases of Influenza to minimize misuse of the surveillance resources.

In this evaluation, we report that; the prevalence of Influenza A (with circulating Influenza A subtype, H3N2) is relatively higher than that of influenza B regardless of the case definition type used, sex, or age category of the individuals under surveillance. Several previous studies have shown similar results from evaluating influenza surveillance [4, 6, 7, 9]. This suggests the need for routine further characterization of the Influenza A virus detected to identify any new strain capable of causing a pandemic.

From a virologic perspective, the surveillance of Influenza in 2019 permitted the characterization of circulating influenza viruses. It ensured strong links with the WHO Collaborating Centers on Influenza by participating in vaccine strain selection and shipping specimens twice a year, as recommended by WHO. Although the NISSS is currently stable due to the external funding mechanism, this suggests the need to deploy internal funding mechanisms to provide support to the MoH, NIC, and sentinel sites. Subsequently, this will create a sense of ownership of the system among the stakeholders and enable the consistently good performance of the NISSS in Tanzania.

## Conclusion

The system performed overall satisfactorily in compliance with its objectives and generated accurate data with an average performance of 100%, which entails a realistic estimation of the burden of influenza disease in the country. As a result, it will assist the country in prioritizing responses to disease outbreaks and even planning for future vaccine development. The complexity of the system contributed to its data inconsistency. Better use of the available data could be made to inform and promote preventive measures, especially among the most vulnerable population (i.e., children aged < 5 years). Increasing sentinel sites could be made to maximize coverage and level of system representativeness.

## Limitations of the study

The sentinel sites used manual registers for the registration of surveillance individuals, while NIC uses an electronic system (Disa*Lab). Since these are two different systems, they may have contributed to the high data discrepancy between NIC and sentinel sites.

## Supporting information

**S1 Fig. Data flow of the National Influenza Sentinel Surveillance System in Tanzania.**
(TIF)

**S1 File. Data collection tool for the evaluation of the National Influenza Surveillance System in Tanzania.**
(PDF)

**S2 File. Updated guidelines for evaluating public health surveillance systems.**
(PDF)

**S3 File. Verification of the number of samples sent to NIC by using registers from sentinel sites and Laboratory information system (LIS).**
(PDF)

**S4 File. Data set for the ILI and SARI patients enrolled in the NISSS in Tanzania, 2019.**
(ZIP)

## Acknowledgments

Sincere appreciation to the National Public Health Laboratory for technical support. The authors also wish to acknowledge the WHO-emergency preparedness and response, Dar es Salaam, Tanzania, as well as SBNH, DRRH, and MRRH, for their technical support during the data collection period.

## Author Contributions

**Conceptualization:** Vulstan James Shedura, Doreen Kamori.

**Data curation:** Vulstan James Shedura, Salum Kassim Nyanga.

**Formal analysis:** Vulstan James Shedura, Ally Kassim Hussein, Doreen Kamori.

**Methodology:** Vulstan James Shedura.

**Software:** Vulstan James Shedura.

**Supervision:** Geofrey Joseph Mchau.

**Writing – original draft:** Vulstan James Shedura, Doreen Kamori, Geofrey Joseph Mchau.

**Writing – review & editing:** Doreen Kamori, Geofrey Joseph Mchau.

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
