## [Decision Letter · Decision Letter 0]

4 Aug 2022

PONE-D-22-16315Evaluation of the influenza-like illness sentinel surveillance system: A national perspective in Tanzania from January to December 2019PLOS ONE

Dear Dr. Shedura,

Thank you for submitting your manuscript to PLOS ONE. After careful consideration, we feel that it has merit but does not fully meet PLOS ONE’s publication criteria as it currently stands. Therefore, we invite you to submit a revised version of the manuscript that addresses the points raised during the review process.

Please pay particular attention to the structural and textual points raised by the reviewers 

We look forward to receiving your revised manuscript.

Kind regards,

Clement Adebajo Meseko, DVM, PhD

Academic Editor

PLOS ONE

Journal Requirements:

a) Did participants provide their written or verbal informed consent to participate in this study?

4. We note that Figure 1 in your submission contain map/satellite images which may be copyrighted. All PLOS content is published under the Creative Commons Attribution License (CC BY 4.0), which means that the manuscript, images, and Supporting Information files will be freely available online, and any third party is permitted to access, download, copy, distribute, and use these materials in any way, even commercially, with proper attribution. For these reasons, we cannot publish previously copyrighted maps or satellite images created using proprietary data, such as Google software (Google Maps, Street View, and Earth). For more information, see our copyright guidelines: http://journals.plos.org/plosone/s/licenses-and-copyright.

a) You may seek permission from the original copyright holder of Figure 1 to publish the content specifically under the CC BY 4.0 license.  

5. We note you have included a table to which you do not refer in the text of your manuscript. Please ensure that you refer to Table 4 in your text; if accepted, production will need this reference to link the reader to the Table.

Reviewers' comments:

Reviewer's Responses to Questions

**Comments to the Author**

1. Is the manuscript technically sound, and do the data support the conclusions?

Reviewer #1: Yes

Reviewer #2: Partly

2. Has the statistical analysis been performed appropriately and rigorously? 

Reviewer #1: Yes

Reviewer #2: N/A

3. Have the authors made all data underlying the findings in their manuscript fully available?

Reviewer #1: Yes

Reviewer #2: No

4. Is the manuscript presented in an intelligible fashion and written in standard English?

Reviewer #1: Yes

Reviewer #2: No

5. Review Comments to the Author

Reviewer #1: The World Health Organization (WHO) estimates that seasonal influenza causes a great burden of disease and encourages all nations to establish a Surveillance system that allows for its monitorization and control. In Tanzania, the National Influenza Sentinel Surveillance System (NISSS) was initiated in 2008 and has not since been formally assessed.

Find attached document with comments and suggested changes.

Reviewer #2: In “Evaluation of the influenza-like illness sentinel surveillance system: A national perspective in Tanzania from January to December 2019”, the authors aims at assessing the performances of the sentinel influenza surveillance system implemented in Tanzania. This is a very important topic. Regular evaluation of the existing sentinel surveillance systems allows to continually improve them, and this study is worth publishing. However, the manuscript is not clear enough and not always presented in a logical way. Some information are not given in the introduction or in the methods section to fully understand the message that the authors want to convey.

Avoid the term ‘client’ when talking about ‘patient’ or ‘case’.

Use of semi-comma should be checked throughout the manuscript.

L83: do the authors mean GISRS network?

In the introduction, it would useful to give readers some information about influenza in Tanzania to better understand the epidemic situation. Is influenza seasonal with epidemic peaks or circulating all year-round at lower level?

L178-181 : are we talking about the same three sites? If so, this is a repetition of the selection criteria already mentioned.

The methods section seems to have most of the required information, but the structure could be improved to avoid repetitions and to make things clearer.

There seems to be a mix between the presentation of the results of the surveillance (incidence of influenza and so on) and the evaluation of the surveillance system itself (which would be more about completion of data, performance scores and so on).

The evaluation process is not completely clear. The performances scores and the items taken into consideration as well as the way they were evaluated should be clearly presented. See below about table 4.

Since there is a presentation of the surveillance results (from L212) including the proportion of influenza positive, the laboratory methods used should be detailed in the methods section (PCR? RAT?).

Evaluation of the surveillance system should include the analysis of the proportion of ILI/SARI cases enrolled by each site with regards to the capture population or general data on the site activity (ex for an hospital: total number of patients admitted for any reason). It should also focus on analyzing the age composition of the cases. There is a high proportion of children and the median age very low. This should be analyzed with regards to the population in the region around each site, in order to analyze why the system performs less for adults, for example, and thus be able to propose options to improve the representativeness. Or to discuss this point with regards to cultural practices or socio-economic elements.

Table 1: what is a ‘non-human’ sample?

L218-220: Numbers do not match with Table 1. The proportions are wrong and not calculated on the number of ILI or SARI cases, but on the number of influenza A positives. Which does not really make sense (influenza B among influenza A).

TAT could be one of the criteria to use for the evaluation of the surveillance system. It should probably be analyzed further per site, with regards to the distance to the NIC (if all analyses were performed at the NIC, which is not clear). Here also, so that options to improve TAT could be formulated. This should also be discussed with regards to the goal of the surveillance: is it to provide a diagnosis that will be used in the patient management? Or is it only for surveillance purposes? Not respecting the TAT doesn’t have the same consequence.

English should be checked throughout the manuscript. There are some clumsy sentences. L261-263 is an example. The first sentence starts with ‘although’, which calls for a main clause in the sentence. This main clause is probably the following sentence starting with ‘however’.

L263-264 and Table 2: here is an example showing that not enough information was given in the methods section about the evaluation criteria. Footnote in table 2 gives an indication that 80 samples per month were requested and indeed, presenting the average number of sample per month is an indicator of performance. However, it should perhaps be discussed based on the epidemiological situation and thus, it might vary during the year/season. The proportion of samples taken out of the number of SARI cases is another interesting indicator which calls for more explanation and discussion. It is not clear if the sentinel sites were requested to perform an exhaustive enrollment of ILI and SARI cases or not. It is not clear also how the authors identified the SARI cases when they audited the 3 sentinel sites: what did they use? Here also, follow-up information is needed for the sites that did not reach the target to understand the reasons and be able to propose options for improvement.

The part on Data Quality mixes methods and results. The methods, describing the indicators used for the evaluation and how they were assessed should be presented before and in a clear way.

L330-333: not clear what the authors are talking about.

L333-334: detection of SARS-CoV-2 in 2019? When? How? The first report of SARS-CoV-2 is from December 2019 and specific RT-qPCR assays were available only in January 2020. How did the authors manage to detect SARS-CoV-2?

Statements in the part on representativeness are not supported by data. Given the low numbers of older adults and their susceptibility to influenza, the proportion of older adults seems low and it is not clear whether or not it matches with the population composition.

L368: the definition of Positive Predictive Value is false. The authors are talking about positivity rate. It is not clear how the authors could calculate a true PPV with the data from this study.

L371-373: not clear. Enrollment being based on a clinical definition, it does not mean that all cases are truly infected by influenza viruses. Other pathogens can lead to a similar pattern of symptoms. The use of ILI/SARI definition is only giving a hint of the situation, but only virological testing can confirm the etiology. There is a complementary of the syndromic and virological surveillances. Which confirms that information on the epidemiological context of influenza in Tanzania is missing for the reader to properly evaluate the results.

Table 4 comes far too late. Elements of this table should have been described before in the methods section.

Part of risk factors associated with influenza has nothing to do with the evaluation of the surveillance system. As already mentioned, there is a confusion between the results of the surveillance and its evaluation. The authors states that they are interested in the evaluation of the system, not in the results. The authors must make a choice about what they want to focus on.

Figure 1 is not acceptable for publication.

In general, quality of the figure should be improved.

6. PLOS authors have the option to publish the peer review history of their article (what does this mean?). If published, this will include your full peer review and any attached files.

Reviewer #1: No

Reviewer #2: No

---

## [Author Response · Author response to Decision Letter 0]

17 Sep 2022

Rebuttal letter 

Response to reviewers

We thank the editor and reviewers for evaluating our manuscript. We provide line-by-line responses to comments raised. Please, follow our responses in a green mark. to every comment/question in yellow mark. We copied the review decision from the submission menu of the editorial manager. We used the same letter to write our responses. 

PONE-D-22-16315

Evaluation of the influenza-like illness sentinel surveillance system: A national perspective in Tanzania from January to December 2019.

PLOS ONE

Dear editor,

Pls find the modified manuscript. We have incorporated all suggestions recommended by the reviewers. Following is the point-wise reply to the reviewers' comments.

With best regards,

Mr. Shedura

Reply to comments

Comment:

Dear Dr. Shedura,

Thank you for submitting your manuscript to PLOS ONE. After careful consideration, we feel that it has merit but does not fully meet PLOS ONE’s publication criteria as it currently stands. Therefore, we invite you to submit a revised version of the manuscript that addresses the points raised during the review process.

Please pay particular attention to the structural and textual points raised by the reviewers. Please submit your revised manuscript by Sep 18 2022 11:59PM. If you will need more time than this to complete your revisions, please reply to this message or contact the journal office at plosone@plos.org. Response: Thank you for the comments.

Comment:

Response: The rebuttal letter has been uploaded as a separate file labeled 'Response to Reviewers'.

Comment:

Response: The revised marked copy of the manuscript that highlights changes made has been uploaded as a separate file labeled ‘revised Manuscript with track changes’. 

Comment:

Response: Unmarked manuscript without tracked changes has been uploaded as a separate file labeled ‘Manuscript’.

Comment:

Response: We have made no changes in financial disclosure statement. 

Comment:

Response: We have deposited the laboratory protocols used in protocols.io. We used the updated guidelines for evaluation of the public health surveillance systems, available at https://stacks.cdc.gov/view/cdc/13376.

Comment:

We look forward to receiving your revised manuscript.

Kind regards,

Clement Adebajo Meseko, DVM, PhD

Academic Editor

PLOS ONE

 Response: Thank you for the comments. 

Journal Requirements:

Comment:

Response: We have modified the style of our manuscript as required by Plos One.

Comment:

a) Did participants provide their written or verbal informed consent to participate in this study?

Response: We have amended the ethics statement and made clear that the study did not involve direct contact with participants and therefore it based on the verbal consent obtained from patients prior to enrolment into the surveillance. Furthermore; the study was done within the framework of the Integrated Disease Surveillance and Response matrix implemented by the Tanzania Ministry of Health and therefore did not require a formal review by Ethical Review Committees. 

Comment:

Response: The evaluation did not involve direct contact with participants and therefore it based on the verbal consent obtained from patients prior to enrolment into the surveillance. Furthermore; the study was done within the framework of the Integrated Disease Surveillance and Response matrix implemented by the Tanzania Ministry of Health and therefore did not require a formal review by Ethical Review Committees. The consent procedures were conducted and approved as per standard procedures of the NISSS.

Comment:

Response: The raw data in STATA data sheet has been uploaded.

Comment:

Response: We have revised the cover letter and shared the dataset as required by PLOS ONE.

Comment:

Response: There are no restrictions. The dataset file is being uploaded as required.

Comment:

4. We note that Figure 1 in your submission contain map/satellite images which may be copyrighted. All PLOS content is published under the Creative Commons Attribution License (CC BY 4.0), which means that the manuscript, images, and Supporting Information files will be freely available online, and any third party is permitted to access, download, copy, distribute, and use these materials in any way, even commercially, with proper attribution. For these reasons, we cannot publish previously copyrighted maps or satellite images created using proprietary data, such as Google software (Google Maps, Street View, and Earth). For more information, see our copyright guidelines: http://journals.plos.org/plosone/s/licenses-and-copyright.

a) You may seek permission from the original copyright holder of Figure 1 to publish the content specifically under the CC BY 4.0 license. 

b) If you are unable to obtain permission from the original copyright holder to publish these figures under the CC BY 4.0 license or if the copyright holder’s requirements are incompatible with the CC BY 4.0 license, please either i) remove the figure or ii) supply a replacement figure that complies with the CC BY 4.0 license. Please check copyright information on all replacement figures and update the figure caption with source information. If applicable, please specify in the figure caption text when a figure is similar but not identical to the original image and is therefore for illustrative purposes only.The following resources for replacing copyrighted map figures may be helpful:

 USGS National Map Viewer (public domain): http://viewer.nationalmap.gov/viewer/The Gateway to Astronaut Photography of Earth (publicdomain): http://eol.jsc.nasa.gov/sseop/clickmap/

Maps at the CIA (public domain): https://www.cia.gov/library/publications/the-world-factbook/index.html and https://www.cia.gov/library/publications/cia-maps-publications/index.html NASA Earth Observatory (public domain): http://earthobservatory.nasa.gov/

Landsat: http://landsat.visibleearth.nasa.gov/ USGS EROS (Earth Resources Observatory and Science (EROS) Center) (public domain): http://eros.usgs.gov/#

Response: We have removed the previous Figure 1 on the revised manuscript.

Comment:

5. We note you have included a table to which you do not refer in the text of your manuscript. Please ensure that you refer to Table 4 in your text; if accepted, production will need this reference to link the reader to the Table.

Response: We have cited Table 4 in the text which has now changed to Table 3 in the revised manuscript.

Academic Editor comments

Comment:

Instead of the disease, it would be best to name it as Influenza. Instead of the distribution, to be more general here it might be best to say “estimating the burden of disease caused by Influenza virus….” This is only in Tanzania, correct? This should be stated.

Response: We have replaced the word ‘disease’ by ‘Influenza’. We have stated the name of the country (Tanzania) in which the surveillance system was evaluated.

Comment:

It be more clear to start out with: From March to April 2021, we collected data

Response: We have begun the sentence by the phrase “From March to April 2021”, as suggested. 

Comment:

No references in abstract

Response: We have removed reference from the abstract. 

Comment:

The words “A review of the Influenza Laboratory electronic forms from the Laboratory information system (Disa Lab) showed that” has been stated in the methodology section.

Response: The words “A review of the Influenza Laboratory electronic forms from the Laboratory information system (Disa Lab) showed that” is being removed.

Comment:

What does this TAT relate to? Not mentioned in Methods, Delete from abstract

Response: The Turnaround time (TAT) relates to the timeliness of the Tanzania Influenza surveillance system evaluated. Shorter TAT may lead to timely operations of the system procedures. TAT is one of the timeliness indicators involved in the evaluation of the Influenza surveillance system and we have included it in the method section. This definition has been added in the revised manuscript in page 6

Comment:

Is NPHL- National Public Health Laboratory?

Response: Yes, the abbreviation has been defined in the revised manuscript. 

Comment:

“since then, epidemics have been persisting”. There were epidemics before, only they were not recorded. So, since the instauration of the NISSS.” influenza epidemics have been monitored”

Response: The sentence has been rephrased to “since then, epidemics have been reported and monitored”.

Comment:

The words “National Influenza Sentinel Surveillance System (NISSS)”: It has already been specified, delete (NISSS) or leave and delete National Influenza Sentinel Surveillance System

Response: We have deleted the words “National Influenza Sentinel Surveillance System” in the sentence and replaced them with “NISSS” as suggested. 

Comment:

The words “The NISSS cover the entire population of Tanzania, which was approximated to be 56.32 million by 2019”. This is a sentinel system; how can it be inclusive to the entire population? By definition it would not be a sentinel system then.

Response: We have deleted the sentence. We have made it clear that: The population under NISSS involves patients enrolled from the fourteen sentinel sites located in eight regions of Tanzania. We have listed the fourteen sentinel sites of the NISSS in Tanzania in the methodology part.

Comment:

Decide whether the participants are to be patients or clients throughout the paper. To me the first option is more adequate

Response: We have decided to use Patients instead of Clients throughout the paper

Comment:

Are these IQR correct? (In the age variable, table 1)

Response: Yes, the IQR are correct. We have shown them clear both median and their interquartile ranges (IQR).

Comment:

About “Non-human sample (Failed results). Can you explain this?

Response: Non-human samples are samples contaminated with PCR inhibitors resulting for their detection as “no-human sample” by RT PCR Machine, and therefore, the influenza virus status in the sample can’t be obtained. We have changed the word to “Failed samples” instead of “Non-human sample”. This clarification has been added in the revised manuscript in page 9 

Comment:

“System attributes”-Consider reducing the description of these items, score results of which are presented on Table 5. And include the information in the discussion section 

Response: We have reduced the description of the individual system attributes and included some information in methodology and discussion sections.

Comment:

In system attributes description “Simplicity”. These 2 sentences are not clear

Response: The sentences have been rephrased and made clear.

Comment:

“Table 3”: Consider if really relevant to include in the text (maybe as supplementary material) 

Response: Table 3 in the text has been shifted to supplementary material.

Comment:

“Uncharacterized influenza A viruses into subtypes”. This is a limitation to which of the considered items?

Response:

We have removed the sentence since it is irrelevant to the objectives of the study.

Response to Reviewers’ comments

Reviewers' comments:

Reviewer's Responses to Questions

Comments to the Author

Comment:

1. Is the manuscript technically sound, and do the data support the conclusions?

Reviewer #1: Yes

Reviewer #2: Partly

Response: Thank you for the comments

Comment:

2. Has the statistical analysis been performed appropriately and rigorously?

Reviewer #1: Yes

Reviewer #2: N/A

Response: Thank you for the comments

Comment:

3. Have the authors made all data underlying the findings in their manuscript fully available?

Reviewer #1: Yes

Reviewer #2: No

Response: Thank you for the comments. The STATA file has been uploaded as a supporting information.

Comment:

4. Is the manuscript presented in an intelligible fashion and written in standard English?

Reviewer #1: Yes

Reviewer #2: No

Response: We have used the standard English in the revised manuscript.

5. Review Comments to the Author

Comment:

Reviewer #1: The World Health Organization (WHO) estimates that seasonal influenza causes a great burden of disease and encourages all nations to establish a Surveillance system that allows for its monitorization and control. In Tanzania, the National Influenza Sentinel Surveillance System (NISSS) was initiated in 2008 and has not since been formally assessed.

Response: The first National Influenza Sentinel Surveillance System (NISSS) in Tanzania was conducted in 2010 (30 months after the initiation of the surveillance system), and revealed the prevalence of seasonal influenza in Tanzania of 8.0%. We have included this information in the introduction part. 

Comment:

Find attached document with comments and suggested changes.

Response: Thank you for the comments and suggestions. We have reviewed the comments and addressed them. 

Comment:

Reviewer #2: In “Evaluation of the influenza-like illness sentinel surveillance system: A national perspective in Tanzania from January to December 2019”, the authors aims at assessing the performances of the sentinel influenza surveillance system implemented in Tanzania. This is a very important topic. Regular evaluation of the existing sentinel surveillance systems allows to continually improve them, and this study is worth publishing. However, the manuscript is not clear enough and not always presented in a logical way. Some information is not given in the introduction or in the methods section to fully understand the message that the authors want to convey.

Response: We have revised the manuscript flow with inclusion of the detailed PCR technique procedures used for the study in the methodology section. 

Comment:

Avoid the term ‘client’ when talking about ‘patient’ or ‘case’.

Response: We have used the term ‘patient’ instead of ‘client’ throughout the manuscript.

Comment:

Use of semi-comma should be checked throughout the manuscript.

Response: We have checked-up the use of semi-comma throughout the manuscript.

Comment:

L83: do the authors mean GISRS network?

In the introduction, it would useful to give readers some information about influenza in Tanzania to better understand the epidemic situation. Is influenza seasonal with epidemic peaks or circulating all year-round at lower level?

Response: Yes, we mean ‘Global Influenza Surveillance and Response System (GISRS) Network’. We have corrected the spelling errors. Furthermore; we have included the ‘epidemic situation’ and ‘seasonality of influenza’ in Tanzania in the introduction part. 

Comment:

L178-181: are we talking about the same three sites? If so, this is a repetition of the selection criteria already mentioned.

Response:

Thank you for the comment. Yes, we we’re describing the same sites. We have removed them to avoid duplication.

Comment:

The methods section seems to have most of the required information, but the structure could be improved to avoid repetitions and to make things clearer.

There seems to be a mix between the presentation of the results of the surveillance (incidence of influenza and so on) and the evaluation of the surveillance system itself (which would be more about completion of data, performance scores and so on).

The evaluation process is not completely clear. The performances scores and the items taken into consideration as well as the way they were evaluated should be clearly presented. See below about table 4.

Response: We conducted the evaluation of the National Influenza Sentinel Surveillance System (NISSS) in Tanzania, using the CDC updated protocol. Each system attribute was evaluated depending on the indicators shown in Table 4, and as described in the National Influenza Surveillance System protocol of World Health Organization (WHO). The performance scores were set and agreed by the National Influenza Technical working group (NITWG) that includes; public health specialists, Health officers and epidemiologists of the Ministry of Health of Tanzania. 

Comment:

Since there is a presentation of the surveillance results (from L212) including the proportion of influenza positive, the laboratory methods used should be detailed in the methods section (PCR? RAT?).

Evaluation of the surveillance system should include the analysis of the proportion of ILI/SARI cases enrolled by each site with regards to the capture population or general data on the site activity (ex for a hospital: total number of patients admitted for any reason). It should also focus on analyzing the age composition of the cases. There is a high proportion of children and the median age very low. This should be analyzed with regards to the population in the region around each site, in order to analyze why the system performs less for adults, for example, and thus be able to propose options to improve the representativeness. Or to discuss this point with regards to cultural practices or socio-economic elements.

Response: We have added the detailed information on PCR technique and procedures used in the methodology part (Laboratory procedures). The description of the age specific ILI/SARI cases is shown in Table 1. The description of number of patients admitted for ILI/SARI is shown in Table 2.

Comment:

Table 1: what is a ‘non-human’ sample?

Response: Non-human samples are samples contaminated with PCR inhibitors resulting for their detection as “no-human sample” by RT PCR Machine, and therefore, the influenza virus status in the sample can’t be obtained. We have changed the word to “Failed samples” instead of “Non-human sample”. We have shown this in Table 1.

Comment:

L218-220: Numbers do not match with Table 1. The proportions are wrong and not calculated on the number of ILI or SARI cases, but on the number of influenza A positives. Which does not really make sense (influenza B among influenza A).

TAT could be one of the criteria to use for the evaluation of the surveillance system. It should probably be analyzed further per site, with regards to the distance to the NIC (if all analyses were performed at the NIC, which is not clear). Here also, so that options to improve TAT could be formulated. This should also be discussed with regards to the goal of the surveillance: is it to provide a diagnosis that will be used in the patient management? Or is it only for surveillance purposes? Not respecting the TAT doesn’t have the same consequence.

Response: Thank you for the comments. We have corrected the proportions in the revised version of the manuscript. Furthermore; we have added analysis of the Sentinel sites -specific TAT as shown in the current Figure 4. TAT is both for providing prompt diagnosis to patients under surveillance and for ensuring timely operation of the surveillance activities. We have added TAT information in the methodology and discussion sections. 

Comment:

English should be checked throughout the manuscript. There are some clumsy sentences. L261-263 is an example. The first sentence starts with ‘although’, which calls for a main clause in the sentence. This main clause is probably the following sentence starting with ‘however’.

Response: We have reviewed English and corrected grammatical errors throughout the manuscript.

Comment:

L263-264 and Table 2: here is an example showing that not enough information was given in the methods section about the evaluation criteria. Footnote in table 2 gives an indication that 80 samples per month were requested and indeed, presenting the average number of samples per month is an indicator of performance. However, it should perhaps be discussed based on the epidemiological situation and thus, it might vary during the year/season.

Response: The evaluation criteria has been added in the methodology section. The established average of samples to be collected per site is 80 samples per month in each sentinel site regardless of the epidemiological changes, according to the established protocol for the evaluation of National influenza sentinel surveillance system in Tanzania. 

Comment:

The proportion of samples taken out of the number of SARI cases is another interesting indicator which calls for more explanation and discussion. It is not clear if the sentinel sites were requested to perform an exhaustive enrollment of ILI and SARI cases or not. It is not clear also how the authors identified the SARI cases when they audited the 3 sentinel sites: what did they use? Here also, follow-up information is needed for the sites that did not reach the target to understand the reasons and be able to propose options for improvement.

Response: The sentinel sites were not requested to perform an exhaustive enrollment of ILI and SARI cases. The sentinel sites enroll patients who meet the standard case definition for either ILI or SARI. The standard case definitions used (ILI and SARI) are described in the methodology section in page 4 in the inclusion criteria. 

Comment:

The part on Data Quality mixes methods and results. The methods, describing the indicators used for the evaluation and how they were assessed should be presented before and in a clear way.

Response: Thank you for the comment on this. We have reviewed the explanation of data quality and separated the results from the methodology information. We have added in the methodology part a description of the indicators used to assess the system attributes of the National Influenza Sentinel surveillance system in Tanzania.

Comment:

L330-333: not clear what the authors are talking about.

L333-334: detection of SARS-CoV-2 in 2019? When? How? The first report of SARS-CoV-2 is from December 2019 and specific RT-qPCR assays were available only in January 2020. How did the authors manage to detect SARS-CoV-2?

Response: Thank you for your insight on this. In L330-334: we meant that NISSS through the established protocol, allows testing of other respiratory viral infections including SARS-CoV-2, and thus not limited to Influenza virus alone. However; we have removed this part from the revised version of the manuscript for more clarity. 

Comment:

Statements in the part on representativeness are not supported by data. Given the low numbers of older adults and their susceptibility to influenza, the proportion of older adults seems low and it is not clear whether or not it matches with the population composition.

Response: Sentinel sites were chosen from 8 regions based on population density climatic, and socio diversity in the country so as to facilitate estimation of disease burden in the whole population. Despite the fact that there was unequal distribution of patients of different age, all age category was included in the surveillance and thus being representative. 

Comment:

L368: the definition of Positive Predictive Value is false. The authors are talking about positivity rate. It is not clear how the authors could calculate a true PPV with the data from this study.

Response: Thank you for the comments. According to the updated guidelines for evaluating public health surveillance systems available at https://stacks.cdc.gov/view/cdc/13376.(page 18-20), defines Predictive value positive (PVP) of the surveillance system as the proportion of reported cases that actually have the health-related event under surveillance. For this context, the health-related event under surveillance is Influenza disease. Furthermore; from the cited guideline, the calculation of the PVP is done by using a formula: True positive/ (True positive + False negative). The “True positive” of the surveillance system are those patients who met the standard case definition of either SARI or ILI, depending on their clinical presentations, and therefore enrolled in to the surveillance system, and their PCR results showed positive results of the influenza Virus A or Influenza Virus B. So, in this study such patients were 373 in number. However; the other category of “False positive” of the surveillance system are those patients who met the standard case definitions (SARI/ILI), enrolled in to the surveillance system but their PCR results showed no detection of Influenza virus. For this study such patients were 1348 in number. Therefore; according to the PVP formular; True positive/ (True positive + False negative); PVP=373/ (373+1348) =21.7%. In summary; we calculated the PVP of the Influenza surveillance system and not PVP of the influenza test. We have corrected some errors during the calculation in the revised manuscript. 

Comment:

L371-373: not clear. Enrollment being based on a clinical definition; it does not mean that all cases are truly infected by influenza viruses. Other pathogens can lead to a similar pattern of symptoms. The use of ILI/SARI definition is only giving a hint of the situation, but only virological testing can confirm the etiology. There is a complementary of the syndromic and virological surveillances. Which confirms that information on the epidemiological context of influenza in Tanzania is missing for the reader to properly evaluate the results.

Response: Thank you for the comments. The National Influenza Sentinel Surveillance System in Tanzania is flexible, and allows the investigation of other viral respiratory infections apart from Influenza, which may show similar clinical presentation during the time of enrolment. We have described the epidemiological trend of influenza in Tanzania at the introduction part of the revised manuscript in page 2. 

Comment:

Table 4 comes far too late. Elements of this table should have been described before in the methods section.

Part of risk factors associated with influenza has nothing to do with the evaluation of the surveillance system. As already mentioned, there is a confusion between the results of the surveillance and its evaluation. The authors states that they are interested in the evaluation of the system, not in the results. The authors must make a choice about what they want to focus on.

Response: Thank you for this comment. We have added description of the evaluation elements in the methodology section. We mainly aimed to evaluate the surveillance system, and therefore, we have removed the part of risk factors in the revised manuscript. 

Comment:

Figure 1 is not acceptable for publication.

In general, quality of the figure should be improved.

Response: Thank you for the comment. We have removed Figure 1 and listed down the sentinel sites of NISSS in the methodology part. The Figure 4 in the revised manuscript involves the description of TAT coverage in each sentinel sites of the NISSS in Tanzania as suggested. 

Comment:

6. PLOS authors have the option to publish the peer review history of their article (what does this mean?). If published, this will include your full peer review and any attached files. Do you want your identity to be public for this peer review? For information about this choice, including consent withdrawal, please see our Privacy Policy.

Reviewer #1: No

Reviewer #2: No

Response: Thank you for the comments. 

 Comment:

 Response: Thank you.

NB: We have made some changes in the order of figures, tables and supporting information.

Figures:

Fig 1 removed, Fig 2 has changed to Fig 1, Fig 3 has changed to Fig 2, Fig 4 has changed to Fig 3 and the new Fig 4 added. 

Tables:

 Tables, 3 and 5 are removed and table 4 has changed to Table 3. 

Supporting information:

 We have added S3_File and S4_ File.

---

## [Decision Letter · Decision Letter 1]

13 Dec 2022

PONE-D-22-16315R1Evaluation of the influenza-like illness sentinel surveillance system: A national perspective in Tanzania from January to December 2019PLOS ONE

Dear Dr. Shedura,

Thank you for submitting your manuscript to PLOS ONE. After careful consideration, we feel that it has merit but does not fully meet PLOS ONE’s publication criteria as it currently stands. Therefore, we invite you to submit a revised version of the manuscript that addresses the points raised during the review process.

We look forward to receiving your revised manuscript.

Kind regards,

Clement Adebajo Meseko, DVM, PhD

Academic Editor

PLOS ONE

Journal Requirements:

Additional Editor Comments:

The manuscript is an important public health surveillance data for influenza and other respiratory viruses but still in need of clarity and improvement including grammar. Note the comments by both reviewers that subtype H5 is not a seasonal influenza and it is important to clarify if this is a valid detection because of the importance of H5 in global public health and influenza virus reporting to the WHO.

Reviewers' comments:

Reviewer's Responses to Questions

**Comments to the Author**

1. If the authors have adequately addressed your comments raised in a previous round of review and you feel that this manuscript is now acceptable for publication, you may indicate that here to bypass the “Comments to the Author” section, enter your conflict of interest statement in the “Confidential to Editor” section, and submit your "Accept" recommendation.

Reviewer #1: All comments have been addressed

Reviewer #2: (No Response)

2. Is the manuscript technically sound, and do the data support the conclusions?

Reviewer #1: Yes

Reviewer #2: Yes

3. Has the statistical analysis been performed appropriately and rigorously? 

Reviewer #1: Yes

Reviewer #2: No

4. Have the authors made all data underlying the findings in their manuscript fully available?

Reviewer #1: Yes

Reviewer #2: Yes

5. Is the manuscript presented in an intelligible fashion and written in standard English?

Reviewer #1: Yes

Reviewer #2: No

6. Review Comments to the Author

Reviewer #1: Line 132 All specimens positive for influenza A virus were subtyped for seasonal H1(A[H1], H3(A[H3]), H5(A[H5]) and A(H1N1) pdm09

The () are not correctly distributed and H5 is not a seasonal subtype but avian , it should read :All specimens positive for influenza A virus were subtyped for H1, H3, H5 and H1N1pdm09

Yet is this really so? H5 subtyping is carried out in sentinell seasonal influenza surveillance?

Reviewer #2: English needs to be reviewed, especially in modified text (in red in manuscript with corrections). New or rewritten parts need to be checked thoroughly . The sentences are sometimes clumsy.

The description of the evaluation criteria is very useful and enhances greatly the manuscript. The authors should however now work on simplifying the manuscript. Perhaps that the criteria and the results could be extensively presented in a table (i.e. table 3 with additional information if needed), but maybe less extensively in the text.

“clients” is still used several times in the manuscript.

Laboratory procedures: H5 is not considered as a seasonal human influenza viruses.

Predictive Positive Value: it is still not clear how the authors were able to calculate this. This would require to have the number of influenza positive and negative samples among cases (matching the case definitions) and non-cases. The surveillance system aims at enrolling only cases. The authors have thus access only to True Positive and False Negative (what the authors wrongly call False Positive).

The manuscript is really focusing on a major aspect of surveillance: its evaluation. There are not many papers providing such an evaluation, so the manuscript is really important for the scientific community involved in public health surveillance of influenza and other respiratory viruses. It is so important in fact, that it is worth for the authors to take a bit more time to make their manuscript as clear and easy to follow as possible. As such, the content is ok, but the form can be improved in order to avoid too long sentences and to correct the grammar. The clarity of the message will gain from a more synthetic presentation.

7. PLOS authors have the option to publish the peer review history of their article (what does this mean?). If published, this will include your full peer review and any attached files.

Reviewer #1: No

Reviewer #2: No

---

## [Author Response · Author response to Decision Letter 1]

4 Jan 2023

Rebuttal letter 

Response to reviewers

We thank the editor and reviewers for evaluating our manuscript. We provide line-by-line responses to comments raised. Please, follow our responses in a green mark. to every comment/question in yellow mark. We copied the review decision from the submission menu of the editorial manager. We used the same letter to write our responses. 

PONE-D-22-16315R1

Evaluation of the influenza-like illness sentinel surveillance system: A national perspective in Tanzania from January to December 2019.

PLOS ONE

Dear Editor,

Please find the modified manuscript. We have incorporated all suggestions recommended by the reviewers. Following is the point-wise reply to the reviewers' comments.

With best regards,

Mr. Shedura

Reply to comments

Comment:

Dear Dr. Shedura,

Thank you for submitting your manuscript to PLOS ONE. After careful consideration, we feel that it has merit but does not fully meet PLOS ONE’s publication criteria as it currently stands. Therefore, we invite you to submit a revised version of the manuscript that addresses the points raised during the review process.

Comment:

Response: The rebuttal letter has been uploaded as a separate file labeled 'Response to Reviewers'.

Comment:

Response: The revised marked copy of the manuscript that highlights changes made has been uploaded as a separate file labeled ‘revised Manuscript with track changes. 

Comment:

Response: Unmarked manuscript without tracked changes has been uploaded as a separate file labeled ‘Manuscript’.

Comment:

Response: We have made no changes in financial disclosure statement. 

Comment:

Response: We have deposited the laboratory protocols used in protocols.io. We used the updated guidelines for evaluation of the public health surveillance systems, available at https://stacks.cdc.gov/view/cdc/13376.

Comment:

We look forward to receiving your revised manuscript.

Kind regards,

Clement Adebajo Meseko, DVM, PhD

Academic Editor

PLOS ONE

 Response: Thank you for the comments. 

Journal Requirements:

Comment:

 Response: The references have been reviewed. 

Additional Editor Comments:

Comment:

The manuscript is an important public health surveillance data for influenza and other respiratory viruses but still in need of clarity and improvement including grammar. Note the comments by both reviewers that subtype H5 is not a seasonal influenza and it is important to clarify if this is a valid detection because of the importance of H5 in global public health and influenza virus reporting to the WHO.

 Response: It is true that H5 subtype is Avian not seasonal influenza, thank you for this observation. Although it was part of the included investigations, we have decided to exclude it so as to align with the objectives of our study. We have stated that the Influenza subtype involved are H1, H3 and H1N1pdm09. 

Reviewers' comments:

Reviewer's Responses to Questions

Comments to the Author

Comment:

1. If the authors have adequately addressed your comments raised in a previous round of review and you feel that this manuscript is now acceptable for publication, you may indicate that here to bypass the “Comments to the Author” section, enter your conflict of interest statement in the “Confidential to Editor” section, and submit your "Accept" recommendation.

Reviewer #1: All comments have been addressed

Reviewer #2: (No Response)

Response: Thank you for the comments

Comment:

2. Is the manuscript technically sound, and do the data support the conclusions?

Reviewer #1: Yes

Reviewer #2: Yes

Response: Thank you for the comments

Comment:

3. Has the statistical analysis been performed appropriately and rigorously?

Reviewer #1: Yes

Reviewer #2: No

Response: Thank you for the comments

Comment:

4. Have the authors made all data underlying the findings in their manuscript fully available?

Reviewer #1: Yes

Reviewer #2: Yes

Response: Thank you for the comments

Comment:

5. Is the manuscript presented in an intelligible fashion and written in standard English?

Reviewer #1: Yes

Reviewer #2: No

Response: Thank you for the comments. We have corrected all typographical and grammatical errors in the revised manuscript. 

Comment:

6. Review Comments to the Author

Reviewer #1: Line 132 All specimens positive for influenza A virus were subtyped for seasonal H1(A[H1], H3(A[H3]), H5(A[H5]) and A(H1N1) pdm09

The () are not correctly distributed and H5 is not a seasonal subtype but avian, it should read :All specimens positive for influenza A virus were subtyped for H1, H3, H5 and H1N1pdm09

Yet is this really so? H5 subtyping is carried out in sentinel seasonal influenza surveillance?

Response: It is true that H5 subtype is avian not seasonal influenza, thank you for this observation. Although it was part of the included investigations, we have decided to exclude it so as to align with the objective of our study. We have stated that the Influenza subtype involves H1, H3, and H1N1pdm09. 

Comment:

Reviewer #2: English needs to be reviewed, especially in modified text (in red in manuscript with corrections). New or rewritten parts need to be checked thoroughly. The sentences are sometimes clumsy.

The description of the evaluation criteria is very useful and enhances greatly the manuscript. The authors should however now work on simplifying the manuscript. Perhaps that the criteria and the results could be extensively presented in a table (i.e. table 3 with additional information if needed), but maybe less extensively in the text.

Response: Thank you for the comments. We have corrected all typographical and grammatical errors in the revised manuscript. We have simplified the Manuscript in the revised version. 

Comment:

“clients” is still used several times in the manuscript.

Laboratory procedures: H5 is not considered as a seasonal human influenza viruses.

Response: We have removed the word “clients” throughout the manuscript and replaced it with the word “patents”. We have removed the avian influenza subtype (H5) in the revised manuscript to make it clear pertaining to seasonal influenza sentinel surveillance. 

Comment:

Predictive Positive Value: it is still not clear how the authors were able to calculate this. This would require to have the number of influenza positive and negative samples among cases (matching the case definitions) and non-cases. The surveillance system aims at enrolling only cases. The authors have thus access only to True Positive and False Negative (what the authors wrongly call False Positive).

Response: Thank you for the comments. It is true that the surveillance system aims at enrolling only cases, but in influenza surveillance, there are suspected cases and confirmed cases. All cases enrolled in NISSS are suspected cases of influenza that meet either the ILI or SARI case definition. Samples are drawn from all suspected cases for laboratory confirmation and those who will be detected positive are the confirmed cases (true positives) and those who will detect negative after confirmation are the “false positives”. According to the updated guidelines for evaluating public health surveillance systems available at https://stacks.cdc.gov/view/cdc/13376.(pages; 18-20), defines the Predictive value positive (PVP) of the surveillance system as the proportion of reported cases that actually have the health-related event under surveillance. For this context, the health-related event under surveillance is Influenza disease. Furthermore; from the cited guideline, the calculation of the PVP is done by using a formula: True positive/ (True positive + False negative). The “True positive” of the surveillance system are those patients who met the standard case definition of either SARI or ILI, depending on their clinical presentations, and therefore enrolled in to the surveillance system, and their PCR results showed positive results of the influenza Virus A or Influenza Virus B. So, in this study such patients were 373 in number. However; the other category of “False positive” of the surveillance system are those patients who met the standard case definitions (SARI/ILI), enrolled in to the surveillance system but their PCR results showed no detection of Influenza virus. For this study such patients were 1348 in number. Therefore; according to the PVP formular; True positive/ (True positive + False negative); PVP=373/ (373+1348) =21.7%. In summary; we calculated the PVP of the Influenza surveillance system and not PVP of the influenza test. 

Comment:

The manuscript is really focusing on a major aspect of surveillance: its evaluation. There are not many papers providing such an evaluation, so the manuscript is really important for the scientific community involved in public health surveillance of influenza and other respiratory viruses. It is so important in fact, that it is worth for the authors to take a bit more time to make their manuscript as clear and easy to follow as possible. As such, the content is ok, but the form can be improved in order to avoid too long sentences and to correct the grammar. The clarity of the message will gain from a more synthetic presentation.

Response: Thank you for the comments.

Comment:

7. PLOS authors have the option to publish the peer review history of their article (what does this mean?). If published, this will include your full peer review and any attached files. 

Do you want your identity to be public for this peer review? For information about this choice, including consent withdrawal, please see our Privacy Policy.

Reviewer #1: No

Reviewer #2: No

Response: Thank you for the comments.

Comment:

Response: Thank you.

---

## [Editor Report · Decision Letter 2]

1 Mar 2023

Evaluation of the influenza-like illness sentinel surveillance system: A national perspective in Tanzania from January to December 2019

PONE-D-22-16315R2

Dear Dr. Shedura,

We’re pleased to inform you that your manuscript has been judged scientifically suitable for publication and will be formally accepted for publication once it meets all outstanding technical requirements.

Kind regards,

Clement Adebajo Meseko, DVM, PhD

Academic Editor

PLOS ONE
---

## [Editor Report · Acceptance letter]

12 Mar 2023

PONE-D-22-16315R2 

Evaluation of the influenza-like illness sentinel surveillance system: A national perspective in Tanzania from January to December 2019 

Dear Dr. Shedura:

I'm pleased to inform you that your manuscript has been deemed suitable for publication in PLOS ONE. Congratulations! Your manuscript is now with our production department. 

Kind regards, 

on behalf of

Dr. Clement Adebajo Meseko 

Academic Editor

PLOS ONE